# Signatures of hierarchical temporal processing in the mouse visual system

**Lucas Rudelt**[1,2]*, **Daniel González Marx**[1,2], **F. Paul Spitzner**[1,2], **Benjamin Cramer**[3], **Johannes Zierenberg**[1,2], **Viola Priesemann**[1,2,4]*

**1** Max-Planck-Institute for Dynamics and Self-Organization, Göttingen, Germany, **2** Institute for the Dynamics of Complex Systems, University of Göttingen, Göttingen, Germany, **3** Kirchhoff-Institute for Physics, Heidelberg University, Heidelberg, Germany, **4** Bernstein Center for Computational Neuroscience (BCCN), Göttingen, Germany

* lucas.rudelt@ds.mpg.de (LR); viola.priesemann@ds.mpg.de (VP)

**Data Availability Statement:** Experimental data were obtained from https://allensdk.readthedocs.io/en/latest/visual_coding_neuropixels.html. Analysis code is available at https://github.com/

## Abstract

A core challenge for the brain is to process information across various timescales. This could be achieved by a hierarchical organization of temporal processing through intrinsic mechanisms (e.g., recurrent coupling or adaptation), but recent evidence from spike recordings of the rodent visual system seems to conflict with this hypothesis. Here, we used an optimized information-theoretic and classical autocorrelation analysis to show that information- and correlation timescales of spiking activity increase along the anatomical hierarchy of the mouse visual system under visual stimulation, while information-theoretic predictability decreases. Moreover, intrinsic timescales for spontaneous activity displayed a similar hierarchy, whereas the hierarchy of predictability was stimulus-dependent. We could reproduce these observations in a basic recurrent network model with correlated sensory input. Our findings suggest that the rodent visual system employs intrinsic mechanisms to achieve longer integration for higher cortical areas, while simultaneously reducing predictability for an efficient neural code.

## Author summary

How the brain integrates information across different timescales is a fundamental question in neuroscience. Results from primates suggest that higher areas in cortex are specialized on integrating information on longer timescales through stronger network recurrence. However, due to anatomical differences and conflicting empirical evidence, it remains open whether this is a property that is shared among species as a general feature of temporal processing. For example, in rodents, higher cortical areas show an increase in adaptation, which suggests stronger redundancy reduction that might oppose an enhanced temporal integration. Here, we combined an information theoretic analysis with an analysis of correlation timescales and found an increase in information and correlation timescales across the anatomical hierarchy of the mouse visual system. Notably, this upward trend is accompanied by a simultaneous reduction in the predictability of single-neuron spiking, suggesting a decrease in temporal redundancy. We could reproduce these

Priesemann-Group/mouse_visual_timescales, and pre-processed data as well as simulation data are available at https://gin.g-node.org/pspitzner/mouse_visual_timescales.

**Funding:** L.R. was supported by the Deutsche Forschungsgemeinschaft (DFG, German Research Foundation) as part of the SPP 2205 - project number 430157073. F.P.S. and V.P. acknowledge funding from the DFG as part of the SFB 1528 "Cognition of Interaction". J.Z. and V.P. acknowledge funding from the DFG under Germany's Excellence Strategy - EXC 2067/1 (MBExC). All authors received support from the Max Planck Society. The funders had no role in study design, data collection and analysis, decision to publish, or preparation of the manuscript.

**Competing interests:** The authors have declared that no competing interests exist.

findings in recurrent network models, which demonstrates that enhanced temporal integration through an increase in recurrence does not necessarily oppose a reduction of redundancy for individual neurons. Together with our empirical findings, this suggests that mouse visual cortex might exploit both, enhanced temporal integration and stronger adaptation, to tune hierarchical temporal processing.

## Introduction

The brain has the ability to seamlessly process and integrate information on vastly different timescales. In primates, past work suggested that this may be supported by two essential features of neocortex: the highly recurrent architecture of cortical networks [1], and an organization of different cortical areas into a temporal processing hierarchy [2–4]. It was found that early sensory areas specialize on fast processing of sensory inputs [5–7], whereas higher (transmodal) areas perform temporal processing with long timescales—combining new information with past information that is maintained over extended periods [8, 9].

This hierarchy is reflected by an increase in the *intrinsic timescale* of neural activity, as measured by the decay rate of autocorrelation [4, 10–12]. In addition, intrinsic timescales were found to be indicative of the specialization for behaviorally relevant computations [13–16]. Finally, there exist gradients in intra-areal properties across the anatomical cortical hierarchy, all of which point to a stronger recurrent coupling for areas specialized on long timescales [17, 18]. In particular, higher cortical areas have an increased dendritic spine density for pyramidal neurons [1, 19], overall excitation-inhibition ratio [20], the expression of related receptor genes [10, 21], gray matter myelination [10, 22], and the strength of functional connectivity [11, 14, 23, 24]. From modelling studies, a stronger recurrence is known to enable stronger and longer-lasting activity fluctuations [15, 25–27], consistent with the increase in timescales for higher areas. Overall, this led to the understanding that in primates, temporal processing is organized hierarchically [3], and that specializations along this hierarchy are likely governed by differences in recurrent coupling [27, 28]. Yet, it is still open how a temporal hierarchy shaped by recurrence fulfills requirements of neural coding and information processing, and whether it manifests as a general organization principle in mammals.

Here, we investigate mouse visual cortex because it comprises an anatomical and functional sensory processing hierarchy that differs from primate visual cortex in many interesting ways [29]. Visual cortex in mice exhibits clear hierarchical feedforward–feedback projection patterns [30], which are paralleled by differences in the recruitment of inhibitory and excitatory neurons [31], and a functional hierarchy that follows the anatomical hierarchy [32]. In addition, the rodent analog of the ventral stream shows a hierarchy of temporal scales where higher areas encode visual information more persistently [33]. Finally, population codes vary between association and sensory cortices [34], and impairments to cortical frontal (transmodal) areas have a greater impact on evidence accumulation over long timescales than posterior (sensory) areas, which also exhibit shorter activity timescales during evidence accumulation [35]. Thus, the mouse visual cortex also shows many signatures of hierarchical temporal processing.

However, the mouse visual cortex was found to differ substantially from primate cortical organization. First, the sensory processing hierarchy seems to be more shallow [36–38], with overall fewer and more primitive higher areas [39], and multisensory integration at a relatively early stage in processing compared to primates [40]. Second, although there exists evidence for gradients in interneuron numbers and intra-cortical connectivity from sensory to transmodal areas [41], the degree of interareal variation of microstructural properties in mice [42, 43] is

far less pronounced than in the highly differentiated primate cortex [44–47]. This raises the question whether a temporal hierarchy shaped by recurrence also characterizes mice, or whether the strong focus on sensory processing in a more shallow hierarchy requires a different, coding-optimized organization altogether.

To address this question, it is important to note that a coding perspective entails a similar trade-off as the temporal-processing perspective (long integration vs. fast relay): In order to increase the signal-to-noise ratio, *robust coding* requires integrating information over time [48], whereas, for low noise, *efficient coding* of sensory information requires temporal decorrelation to reduce redundancies [49–51]. This trade-off can be characterized using the *predictability R*, which quantifies the proportion of information in current neural spiking that can be predicted from the recent past [52]. This predictable information reflects temporal redundancy, and facilitates, for instance, active information storage (maintaining past input to combine it with present input [53–55]) and associative learning [56]. In addition, the closely related *information timescale* $\tau_R$ gives the timescale over which past information has to be integrated for prediction [52]. Together, predictability and information timescales provide a broad view into the neural code, quantifying both, the amount and the timescale of redundancy in neural spiking.

Here, we will use the information timescale, as well as a correlation timescale, to probe for hierarchical temporal processing in mouse visual cortex. Moreover, we will test whether higher cortical areas show an increase in predictability, in line with robust coding, or a decrease in predictability, in line with efficient coding. Finally, we will compare results between spontaneous activity and natural stimuli, which is important to distinguish between stimulus-induced and intrinsically generated timescales and predictability. Together, these results will clarify whether mouse visual cortex shows signatures of hierarchical temporal processing, and whether these are stimulus-induced, indicating a stronger role of feedforward processing, or rather intrinsically generated, indicating a stronger role of recurrent processing.

## Results

To identify systematic differences in temporal processing between sensory processing stages, we analyzed a dataset from simultaneous Neuropixels recordings of the mouse visual system in vivo [32, 57] (Fig 1C). This dataset contains spike trains collected from *n* = 57 experimental sessions in adult mice under different stimulus conditions, and for thousands of neurons from six cortical areas [primary visual cortex (V1), lateromedial area (LM), anterolateral area (AL), rostrolateral area (RL), anteromedial area (AM) and posteromedial area (PM)] and two thalamic areas [lateral geniculate nucleus (LGN) and lateral posterior nucleus (LP)]. We focused on a stimulus condition from the *Functional Connectivity* experimental sessions, where a (repeated) natural movie was used as stimulus (Fig 1D; see Materials and methods for details on the recording and data selection).

### Timescales and predictability of neural spiking activity

To analyze signatures of temporal processing in the recorded spike trains, we quantify single neuron autocorrelation and predictability (Materials and methods). Autocorrelation $C(T)$ considers only the linear dependence to a single point in time with time lag *T* (Fig 1A). In contrast, predictability $R(T)$ gives the proportion of spiking information $R(T)$ that can be predicted from past spiking in an entire past range *T* (Fig 1B), and thus captures all linear and nonlinear dependencies in that range *T* [52].

Using the autocorrelation, we estimate the correlation timescale of spiking activity, which is computed as the decay time $\tau_C$ of an exponentially decaying autocorrelation (Fig 1A, Materials

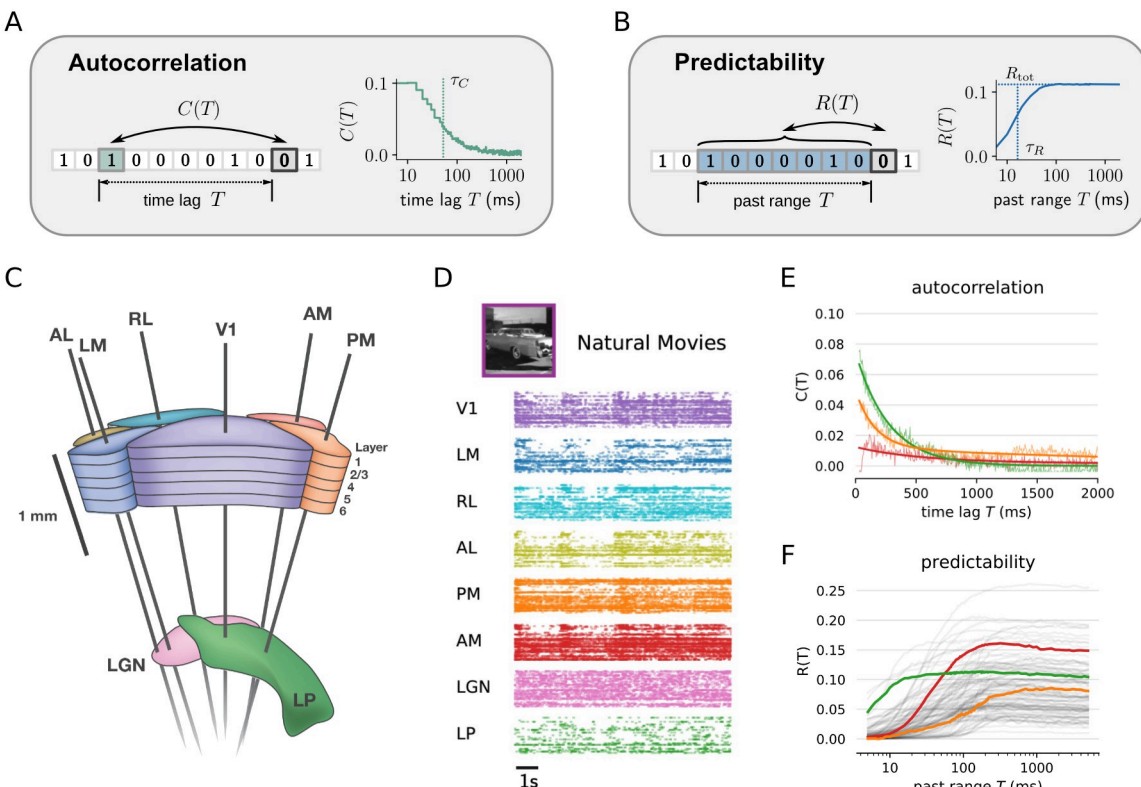

**Fig 1. Timescales and predictability of spiking activity in the mouse visual system. (A)** Autocorrelation $C(T)$ quantifies the correlation between a neuron's spiking activity in two time bins with time lag $T$. Typically, the measured autocorrelation (solid green line) is decaying exponentially with lag $T$ on a characteristic timescale $\tau_C$ (dashed line), here termed correlation timescale. **(B)** Predictability $R(T)$ gives the relative amount of spiking information in a time bin that can be predicted from spikes in all time bins in a past range $T$. $R(T)$ increases monotonously with $T$ until it saturates when reaching the neuron's total predictability $R_{tot}$ (horizontal dashed line), i.e., when adding past bins does not yield additional information. The information timescale $\tau_R$ is defined as the characteristic rise time until $R_{tot}$ is reached (vertical dashed line, c.f. main text). **(C)** Neuropixel-electrodes simultaneously record from up to six visual cortical areas (V1, LM, RL, AL, PM, AM) and two thalamic areas (LGN and LP) (image credit: Allen Institute [58]). **(D)** Example of spiking activity for a random subset of units from different brain areas during stimulation with a natural movie. The box shows a scene from the movie (image adapted from [59]). **(E)** Examples of $C(T)$ for LP, PM and AM (green, orange, red, respectively). $\tau_C$ is estimated by fitting $C(T)$ (thin lines) with an exponential decay (thick lines). **(F)** Examples of $R(T)$, same units as in E.

and methods). It is important to note that the resulting correlation timescale captures temporal correlations from both, intrinsic neural mechanisms and visual stimuli. One way to disentangle these contributions is to work with trial-based data where mean non-stationary inputs can be estimated and removed [4, 33, 60], which gives rise to the so-called intrinsic timescale [4]. Here, we instead compute the correlation timescale for ongoing activity in an entire stimulus block to obtain clean estimates on the single-neuron level. We then systematically compare the estimated timescale under stimulation with a natural movie with the one obtained for spontaneous activity, where the latter reflects the intrinsic timescale [61].

In contrast to the autocorrelation $C(T)$, the predictability $R(T)$ increases monotonously with $T$, because more past information can only increase predictability. From this, we estimate (i) the total predictability $R_{tot}$ of a unit as the value where $R(T)$ saturates for large $T$, as well as (ii) the information timescale $\tau_R$, which can be interpreted as a rise time of the predictability, and indicates a typical timescale on which past activity is informative, i.e. adds to the predictability of current spiking (Fig 1B). Similar to the correlation timescale, the predictability and

information timescale reflect both stimulus-induced and intrinsic effects, and thus will be systematically compared between stimulus conditions.

In the data, we found that single unit autocorrelation is generally well approximated by an exponentially decaying function (Fig 1E and S9 Fig), except for very short time lags, which we accounted for in our fitting procedure (Materials and methods). Moreover, units with higher correlation timescale typically also have higher information timescale (S2 Fig), indicating that the different measures of timescale capture a similar trend in the data. Consistent with previous findings [52], $\tau_R$ is smaller than $\tau_C$, because $R(T)$ only increases for non-redundant past information, i.e. information that could not be read out from a smaller past range $T$. In contrast, the autocorrelation $C(T)$ only considers time-lagged bins, and hence also incorporates redundant contributions [52].

To provide a better intuition of the above measures, we can relate them to simpler statistics of single neuron spiking (S3 Fig). For instance, $\tau_R$ is correlated with the inter-spike-interval (ISI), since a longer ISI implies larger timescales to predict neural spiking. As another example, predictability is correlated with the coefficient of variation (CV), because higher predictability also means a deviation from uncorrelated Poisson spike trains (CV = 1) towards spike trains with higher temporal variability (CV> 1). In contrast, all measures are only weakly correlated with the mean firing rate. Thus, although some relations exist, our chosen measures go beyond simple firing statistics and quantify complementary aspects of the temporal statistics of neural spiking: $\tau_C$ covers linear dependencies, $\tau_R$ captures non-redundant, also non-linear dependencies, and $R_{\text{tot}}$ describes the maximum information that could be predicted over the full past range.

## Timescales and predictability differ between thalamic and cortical visual areas

When comparing the estimated timescales and predictability between different brain areas, we found a significant difference between thalamic and cortical areas (Fig 2A–2C): Units from LGN and LP had much lower timescales $\tau_C$ and $\tau_R$ compared to V1, or higher visual cortical areas. Moreover, predictability $R_{\text{tot}}$ is significantly smaller in thalamus compared to V1 or higher cortical areas. These trends were confirmed with a second, independent dataset (*Brain Observatory 1.1* [32], c.f. S11 Fig). In sum, timescales and predictability in thalamus were smaller than in cortex, which likely reflects the role of thalamus as a relay of information with fast processing, fast forgetting, and temporal decorrelation [62, 63]. In contrast, the higher timescales in cortical areas suggests an enhanced integration of temporal information, which might be supported by the extensive recurrent connectivity in cortex, and longer reverberations of activity in these areas [61]. To better understand how this temporal processing is organized, we next focused on cortical areas.

## Timescales and predictability indicate an organization of temporal processing along the anatomical cortical hierarchy

In mouse visual cortex, the correlation timescale as well as the information timescale and predictability are correlated with the anatomical hierarchy of cortical areas (Fig 2E and 2F). Here, the cortical hierarchy is characterized by the anatomical hierarchy score based on inter-areal feedforward and feedback connectivity from the Allen Mouse Brain Connectivity Atlas [30]. In particular, median timescales increased with hierarchy score (Pearson correlation coefficient $r_P = 0.91$ and $0.87$ for $\tau_C$ and $\tau_R$), indicating longer integration times for higher areas. In contrast, median predictability decreased ($r_P = -0.92$), which indicates less temporal redundancy within individual spike trains in higher cortical areas.

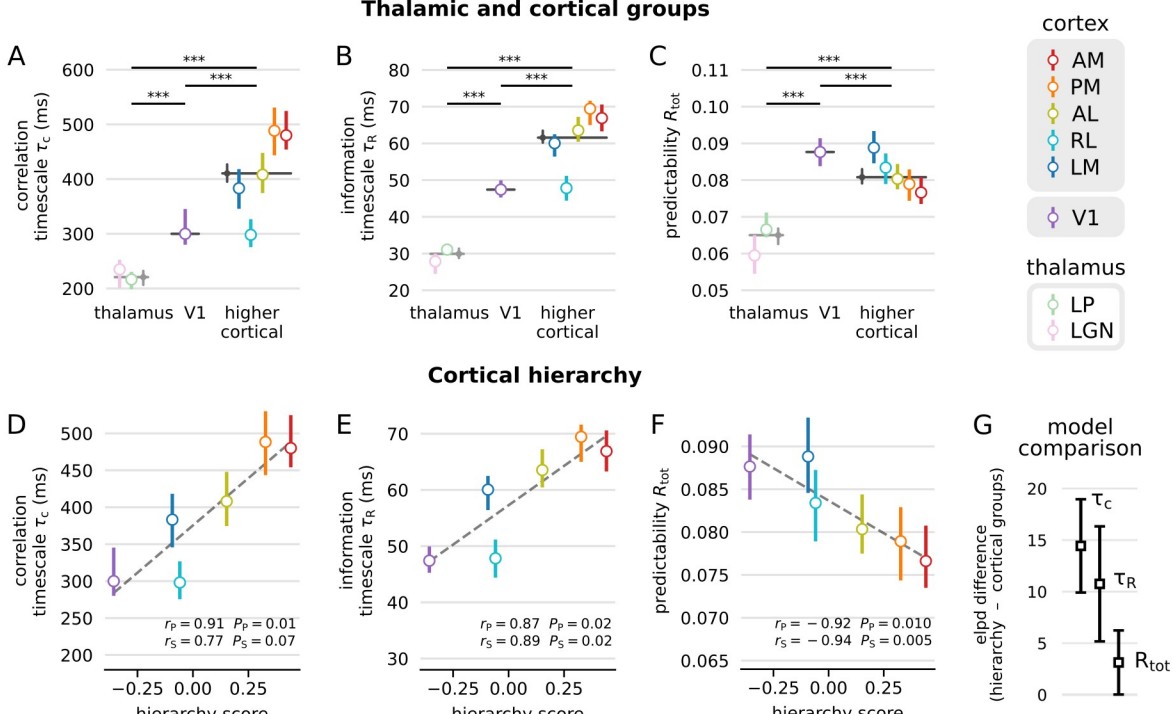

**Fig 2. Timescales and predictability indicate a gradual hierarchy of temporal processing in mouse visual cortex. (A–C)** Hierarchy according to groups of visual areas. Gray markers indicate the median over all sorted units from different groups (thalamus, V1, higher cortical areas), colored dots indicate the median over units for individual areas. Bars indicate 95% confidence intervals on the median obtained by bootstrapping, and p-values for comparisons of different groups were obtained by Mann-Whitney-U tests. **(A)** Correlation timescales $\tau_C$ are lowest for thalamus (areas LGN and LP) compared to primary visual cortex (V1) and higher cortical areas (LM, RL, AL, PM, AM). **(B)** The same trend holds for $\tau_R$, which increases from thalamic areas to V1 to higher cortical areas. **(C)** $R_{tot}$ increases from thalamus to V1, but it is again smaller for higher cortical areas. **(D–F)** Timescales and predictability as a function of anatomical hierarchy score [30, 32]. **(D)** Median $\tau_C$ increases with hierarchy score of the respective cortical areas, which is well approximated by linear regression (dashed line, Pearson and Spearman correlation coefficients and corresponding p-values shown below). **(E)** The same relation holds for $\tau_R$. **(F)** Median $R_{tot}$ in contrast decreases with hierarchy score. **(G)** Model comparison between the model based on cortical groups (V1, higher cortical in A–C), and the model based on linear relationship with hierarchy score (D–F). Shown is the difference in expected log pointwise predictive density (ELPD) from leave-one-out (LOO) cross-validation [64]. Boxes indicate the mean and bars the standard deviation over LOO samples. The cortical hierarchy model has higher predictive power for all measures, but models show more similar performance for $R_{tot}$.

Moreover, we carefully assessed whether the hierarchies of timescales and predictability indeed follow the anatomical hierarchy, or, alternatively, are better described by a grouping into primary and higher, extrastriate cortical areas (V1 vs. higher cortical areas, Fig 2A–2C). This is important, since cortex in rodents is thought to have fewer hierarchical stages, where visual information from V1 is provided more readily (and in parallel) to multimodal interactions or motor outputs [65], which is supported by direct V1 input to essentially all extrastriate (higher cortical) visual areas [66–68]. To this end, we used hierarchical Bayesian modelling (Materials and methods), which allowed us to compare the group and hierarchy hypotheses. Moreover, this allowed us to disentangle differences in temporal processing from trivial differences in average firing rate and visual responsiveness (whether or not units have of a clear receptive field).

For the group hypothesis, we modelled timescales and predictability with a different log mean for each group (V1, higher cortical). For the cortical hierarchy hypothesis we modelled a linear relation between an area's hierarchy score and the median timescales or predictability.

In addition, each model included average firing rate and responsiveness of each unit as predictors.

First, the Bayesian regression confirmed our earlier results: Posteriors over log means of the groups indicated a credible increase in timescales and decrease in predictability for higher cortical areas when compared to V1 (S15 Fig). Moreover, posteriors over the mean slope indicated a credible positive slope for both timescales, and a negative slope for the predictability (S13 Fig).

Second, the Bayesian regression suggested a hierarchical organization over a grouping. When comparing the two models by leave-one-out cross-validation, we found that the hierarchy-score model has more predictive power than the group model for the correlation and information timescale, whereas both models performed more similarly for predictability (Fig 2G). Again, these results were confirmed with the *Brain Observatory* data (S11 and S13 Figs).

In sum, the Bayesian model comparison implied that the increase in timescales is better described by a linear increase with anatomical hierarchy score than a group difference between primary visual cortex and higher cortical areas. This suggests that temporal processing in cortex is organized gradually along the anatomical hierarchy, and not in a two-stage processing architecture.

## Predictability depends on visual stimulation and stimulus selectivity, while timescales are rather invariant

Until now, it is open which mechanisms might underlie the observed hierarchy of temporal processing. One hypothesis is that feedforward processing leads to longer timescales under visual stimulation because of transformation invariant, temporally persistent stimulus representations in higher areas [33, 69–71]. An alternative hypothesis is that stronger recurrence enables longer integration of visual information [15], which is thought to be a cause of long intrinsic timescales in primates [1] To test these hypotheses, we note that feedforward processing predicts overall higher timescales and a more pronounced hierarchy under stimulation with a natural movie (with long stimulus timescales) when compared to spontaneous activity in the absence of a time-varying visual stimulus, where the correlation timescale corresponds to the intrinsic timescale. In contrast, recurrent integration predicts that timescales are rather invariant to the stimulus condition.

When comparing timescales between the previous natural movie condition and spontaneous activity that was recorded while showing the animal a grey screen (Materials and methods), we found that median $\tau_C$ and $\tau_R$ were similar between stimulus conditions (pooled from all units in all cortical areas, Fig 3A and 3B). Moreover, we found that a cortical hierarchy of timescales also exists for spontaneous activity (S12, S13 and S15 Figs). This hierarchy was strikingly similar for the correlation timescale, whereas for the information timescale the correlation with the hierarchy score was smaller under spontaneous activity (S12 Fig) and the Bayesian regression indicated a smaller mean slope (S13 Fig). This suggests that the hierarchy of correlation timescales is little affected by the stimulus condition, while the hierarchy of information timescales becomes more pronounced under visual stimulation.

In contrast to the timescales, median predictability differed between stimuli by $\approx 22\%$ (Fig 3C), which we similarly found when pooling and comparing units for each cortical area separately (S22 Fig). Moreover, when applying the Bayesian regression to predictability for spontaneous activity, we neither found a credible decrease from V1 to higher cortical areas, nor a credible negative slope with the anatomical hierarchy score (S13 and S15 Figs). Thus, while the hierarchy of timescales is found for both stimulus condition, a decrease in predictability shows mainly under a time-varying visual stimulus, where also the overall predictability is larger.

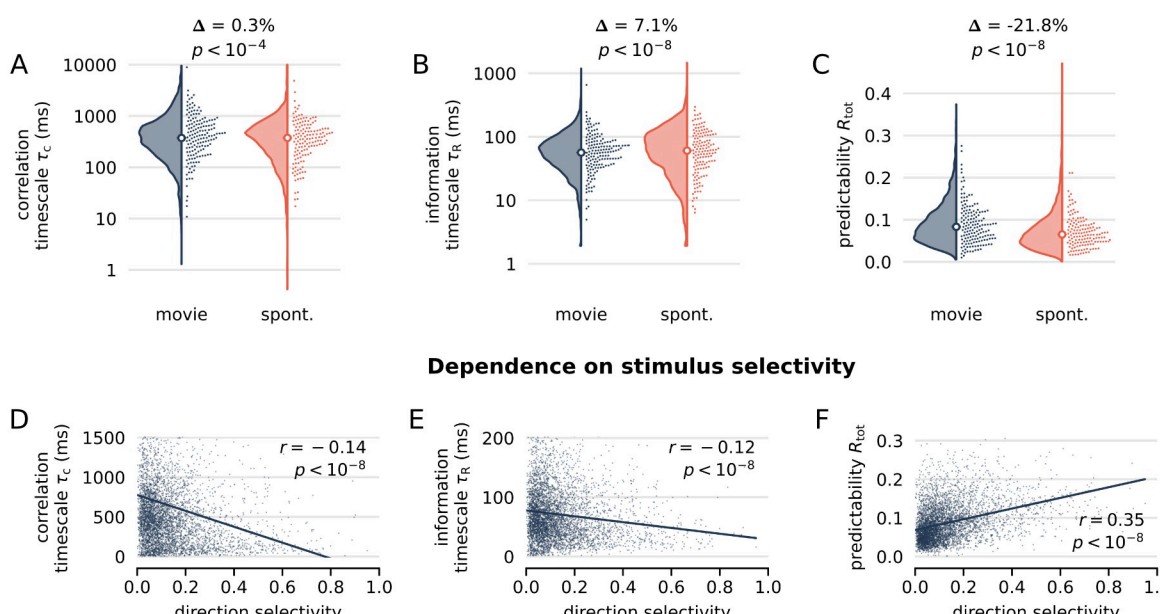

**Fig 3. Predictability depends on visual stimulation and stimulus selectivity, while timescales are rather invariant. (A)** Comparison of correlation timescale $\tau_C$ under stimulation with a natural movie (blue) and spontaneous activity under grey screen illumination (orange), for $N = 4200$ units pooled from all cortical areas. Correlation timescales are very similar between stimulus conditions with difference in medians of $\Delta = 0.3\%$. **(B)** Information timescales $\tau_R$, in contrast, are more broadly distributed for spontaneous activity, with slightly larger difference in medians ($\Delta = 7.1\%$). **(C)** $R_{tot}$ shows the largest difference between stimulus conditions ($\Delta = -21.8\%$). For A–C, p-values are obtained using Wilcoxon signed-rank tests. **(D–F)** Timescales and predictability as a function of direction selectivity, where each dot represents a unit. Timescales and predictability were computed under natural stimulation in the *Brain Observatory 1.1* data set, and direction selectivity was calculated from a separate stimulus condition (drifting gratings). **(D, E)** Timescales $\tau_C$ and $\tau_R$ show a significant yet weak negative correlation with direction selectivity (line shows linear regression, with Pearson correlation $r = -0.14$ and $-0.12$, respectively). **(F)** In contrast, predictability is positively correlated and has a larger correlation coefficient ($r = 0.35$).

To further test the relation between the measures and stimulus encoding, we assessed how timescales and predictability related to stimulus selectivity measures for individual neurons (e.g., direction and image selectivity). To this end, we considered the *Brain Observatory 1.1* data set, which contains recordings under natural movie stimulation and a range of other conditions (drifting gratings, static images) required to quantify stimulus selectivity. From the natural-movie recordings we again estimated timescales and predictability of each unit, and compared them to each unit's direction-selectivity, which was previously computed from drifting-gating recordings [32]. When pooling units across cortical areas, we found that timescales were only weakly and negatively correlated with the direction selectivity ($r = -0.14$ and $r = -0.12$, Fig 3D and 3E), whereas predictability was more strongly and positively correlated ($r = 0.35$, Fig 3F). When repeating the analysis for each cortical area individually, timescales were not significantly correlated with direction-selectivity, but predictability was (S23 Fig). Moreover, we found the same overall dependence as on direction-selectivity also on image-selectivity (S24 and S25 Figs).

Together, these results imply that the hierarchy of timescales is supported by network-intrinsic mechanisms (independent of stimulus conditions or visual coding properties), whereas the hierarchy of predictability is more closely related to visual input. This also suggests that the decrease in predictability along the hierarchy might reflect an active cancellation of predictability in visual stimuli, or is an inherent property of higher-level visual representations (see Discussion).

## Gradients of timescales and predictability are consistent with an increase in recurrence in network models with correlated external input

It has been hypothesized that long correlation timescales are a result of enhanced recurrent activity propagation in cortical networks [1, 25–27], which is in line with the observed stimulus independence of the timescale hierarchy. However, it is not known how higher recurrence affects predictability and information timescales. In particular, one might expect that higher recurrence also leads to more predictable spiking, thus contradicting the observed decrease in predictability for higher areas.

To test this we considered a minimal model of recurrent activity propagation. The branching network [72, 73] consists of sparsely connected units that are either active or silent in each discrete time bin. Activity propagation in the network is governed by the branching parameter $m$, which gives the mean number of units that an active unit activates in the next time step. Moreover, independent of recurrent activations, each unit is activated by temporally uncorrelated external input with mean rate $h$ (Fig 4A). Recurrence in the network is then characterized by the recurrent amplification $a/h = 1/(1 - m)$, which gives the mean number of generated spikes $a$ per external input activation $h$ (c.f. [74]). To test how recurrent amplification $a/h$ affects timescales and predictability, we simulated branching networks consisting of $N = 1000$ units with $k = 10$ outgoing connections each, and estimated $\tau_C$, $\tau_R$ and $R_{\text{tot}}$ using the same procedure as for the experimental data (Materials and methods).

In branching networks, the correlation timescale $\tau_C$ is known to scale linearly with the recurrent amplification $a/h$, which is caused by long-lasting reverberations of activity in the network [15, 25]. We could reproduce this linear increase by fitting $\tau_C$ for individual units in the network, and found a similar increase in $\tau_R$ with $a/h$ (Fig 4B and 4C). Notably, also the predictability $R_{\text{tot}}$ increased with $a/h$, which constitutes an important difference to the experimental data. However, $R_{\text{tot}}$ was more than an order of magnitude smaller than in the experimental data, indicating that a source of predictability is missing in the model (Fig 4D).

Since neurons in cortex receive temporally correlated inputs rather than uncorrelated Poisson noise [51], we replaced the constant external activation rate $h$ for each unit by a fluctuating rate $h(t)$, which was modeled by an Ornstein-Uhlenbeck process with autocorrelation time $\tau_{\text{ext}} = 30$ ms (Fig 4E). As before, the external input is unique for each unit, and thus induces temporal correlations without increasing spatial correlation between units. With temporally correlated input, and intermediate values of recurrent amplification ($0 < a/h \leq 20$), we recovered qualitative agreement between model and experimental data; timescales increased with $a/h$ (Fig 4F and 4G), whereas predictability $R_{\text{tot}}$ decreased until reaching $a/h \approx 20$ (Fig 4H). Moreover, when comparing the two models, timescales share the same overall trend, but predictability differs systematically, and, at small $a/h$, is much higher with correlated than with uncorrelated input (Fig 4H vs. Fig 4D). This is consistent with the observation that the Bayesian analysis that did not find a credible decrease in predictability for spontaneous activity (S13 and S15 Figs), and that predictability is higher under a temporally correlated (movie) stimulus (Fig 3C).

In sum, the combined treatment of recurrence and temporally correlated input suggests that both, an increase in timescales and a decrease in predictability, are consistent with increasing recurrence along the anatomical hierarchy in cortex. Although timescales and predictability in the model are almost an order of magnitude smaller than in the data, the models' simplicity and few mechanistic components enable a systematic explanation of the emergent effect (see Discussion and Fig 4I and 4J). Moreover, we confirmed that the same effect also occurs in a leaky integrate-and-fire (LIF) model with excitatory and inhibitory neurons, which

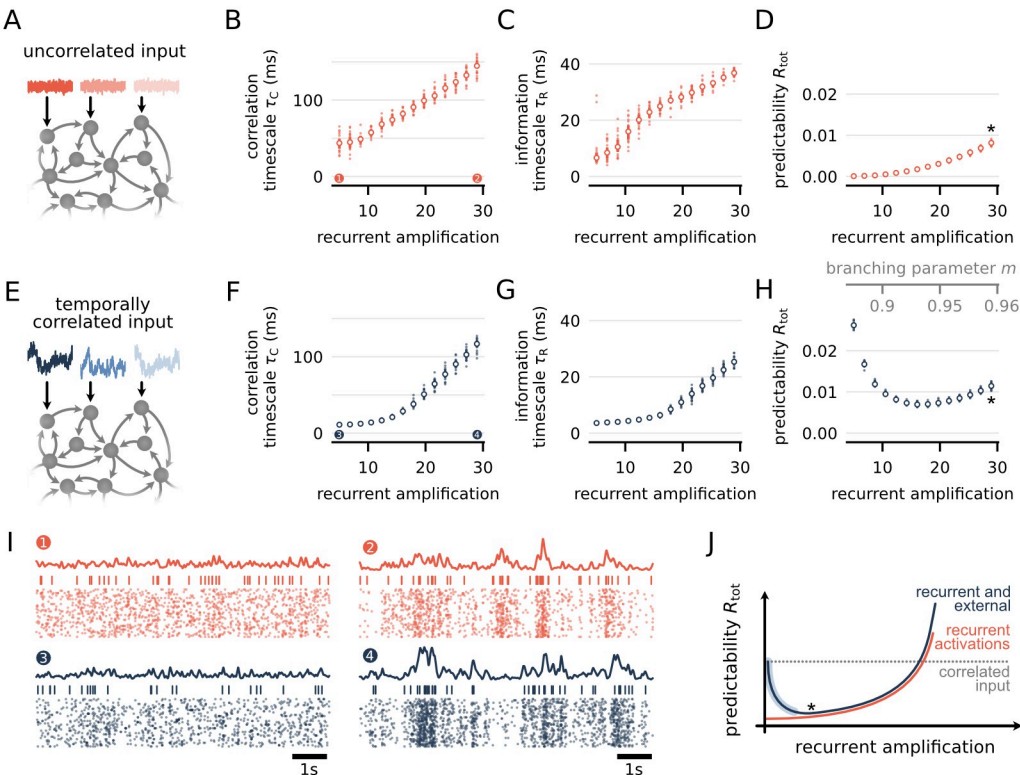

**Fig 4. Gradients of timescales and predictability are consistent with an increase in recurrence in network models with correlated external input. (A)** Schematic of a branching network: Each of the $N = 1000$ units is connected randomly to $k = 10$ other units (arrows). In a given time step, units are either active or inactive, and, if active, can activate each connected neighbour with fixed probability $m/k$ for the next time step. In addition to recurrent activations, each unit gets a temporally uncorrelated external input (red traces). **(B–D)** Timescales and predictability increase with recurrent amplification $a/h = 1/(1 − m)$ in the network, i.e., the mean number of generated spikes per external activation. **(E–H)** Same setup as before, but each unit receives temporally correlated external input that follows an Ohrnstein-Uhlenbeck process with timescale $\tau_{ext} = 30$ms (blue traces). **(F-H)** For correlated input, the timescales behave similarly as before, but predictability is higher and decreases with recurrent amplification until $a/h \approx 20$. All measures are consistently about an order of magnitude smaller than for the experimental data, which we attribute to the simplicity of the model (see main text). Small dots indicate the median over 20 randomly selected units for 10 simulations. Big dots indicate median values over all simulations. **(I)** Example raster plots for uncorrelated (top) and correlated input (bottom), at $a/h = 5$ (left) and $a/h = 30$ (right). Each panel shows the population rate, spiking of a single unit, and population raster of 40 units. **(J)** Sketch of single-unit predictability for different sources of temporal correlations. Recurrent activations yield sizable predictability only for very high recurrent amplification (red line, c.f. panel D). Otherwise, predictability through correlated input is higher (grey dashed line). Hence, in the presence of correlated input, increasing recurrent amplification decreases predictability, because more and more spikes are caused by recurrent activations with lower predictability (blue line, c.f. panel H). When recurrent amplification is high enough (star), most activations are recurrent, and both models give the same behavior.

was implemented on a neuromorphic chip [63, 75] (S26 Fig). In this case, recurrent amplification was adapted through synaptic plasticity, while the membrane dynamics and refractoriness of the LIF neurons provided the additional source of predictability (inputs were temporally uncorrelated). We conclude that the increase in timescales and decrease in predictability is a general effect that occurs when (i) recurrent activity propagation leads to stronger and longer-lasting fluctuations of activity, and (ii) some other source of single-neuron predictability exists, which then has a diminishing effect on single-neuron predictability with increasing recurrence.

## Discussion

Here, we used information theory and an autocorrelation analysis to characterize the temporal statistics of spike trains across the mouse visual system. We found that these statistics differ systematically between processing stages from thalamus to higher cortical areas. In particular, the information and correlation timescales increase along an anatomical hierarchy in cortex, whereas predictability of neural spike trains decreases. Moreover, we found similar gradients and median values of timescales under spontaneous activity, indicating that the observed gradients in timescales are also shaped by network-intrinsic mechanisms like an increase in recurrent coupling that could serve an enhanced temporal integration for higher areas [1, 15, 25, 26, 34]. In contrast, we observed a decrease in predictability mainly under stimulation with a natural movie, which could be the result of an overall higher sensitivity of lower areas to lower-level sensory features with higher predictability [76, 77], or a stronger cancellation of redundant, predictable stimuli, in line with hierarchical predictive and efficient coding [78, 79]. In a basic model of recurrent activity propagation, as well as a plastic leaky integrate-and-fire model on a neuromorphic chip, we verified that information and correlation timescales indeed increase with recurrent coupling, whereas single-neuron predictability can even decrease with recurrence. In sum, our results therefore suggest that enhanced temporal integration in higher areas could be complementary to the observed increase in adaptation along a cortical processing hierarchy [80], where adaptation attenuates responses to redundant, predictable stimuli [81, 82].

### Hierarchy of timescales in rodents

An increase in timescales along the hierarchy was surprising, because the original study by Siegle et al. did not find a hierarchy of intrinsic timescales for spontaneous activity (Extended Data Fig 9 in [32]). However, a later study analysing the same data found a hierarchy of intrinsic timescales [33]—but focused on natural movie stimulation instead of spontaneous activity, and used a different fitting approach [4]. Therefore, we carefully assessed the robustness of our fitting procedure of the correlation timescale $\tau_C$, and found that the discrepancy to the results in [32] is most likely caused by differences in the fitting range (Materials and methods). In particular, we found that when the maximal time lag $T_{max}$ used for fitting is less than 1000 ms, then estimated timescales are generally much smaller (Panel A in S5 Fig), and no correlation between correlation timescales and anatomical hierarchy score is found (S5 and S7 Figs). In contrast, for $T_{max} > 1000$ ms the inferred hierarchy was robust to the exact choice of $T_{max}$ (Panels B and C in S5 Fig). The presented results are further supported by a similar hierarchy of information timescales (Fig 2E), which are estimated with an entirely different approach, and do not rely on fitting an exponential decay rate (Materials and methods).

In primates, a hierarchy of timescales was found along a global hierarchy of cortical areas, spanning different modalities and different levels of cognitive abstraction [4, 83]. Here, for mice, we found a hierarchy of timescales specifically for the visual sensory pathway. This is an important finding, since cortex in rodents is thought to have fewer hierarchical stages, which allows visual information to be provided more readily (and in parallel) to multimodal interactions or motor outputs [65]. This is supported by direct V1 input to essentially all extrastriate (higher cortical) visual areas [66–68], as well as evidence that several extrastriate areas process information related to other sensory modalities [84–86]. At the same time, anatomical feedforward- and feedback-connectivity motifs as well as several functional properties also point to a hierarchical ordering of extrastriate areas [32]. Therefore, we tested whether the observed increase in timescales is gradual (following the anatomical hierarchy), or parallel (reflecting an organization into two stages, V1 vs. extrastriate visual areas). We found that a model with

gradual increase had more predictive power for both, the correlation and information time-scale (Fig 2G, S11 and S12 Figs), suggesting a hierarchical organization of temporal processing in mouse visual cortex.

However, we also found deviations from a single gradient in timescales. In particular, the rostro-lateral area (RL) consistently had the smallest median timescales among extrastriate areas that were more comparable to primary visual cortex, despite its higher anatomical hierarchy score compared to the lateromedial area (LM). This might be due to the functional specialization of the posterior parietal areas (RL, AL, AM) and the posteromedial area (PM), which are thought to process information related to motion and behavioral actions, whereas LM is thought to play a stronger role in object perception, analogous to the dorsal and ventral streams described in other species [87–89]. Thus, our results might also support a grouping of areas into two parallel yet hierarchically organized pathways. This was similarly found in a functional connectivity study that suggested a sensory-to-motor and a transmodal pathway [36].

## Recurrence could shape hierarchy of timescales and predictability

After identifying the gradient of timescales and predictability along the anatomical hierarchy, we uncovered potential mechanisms behind it in an easy-to-interpret model of recurrent activity propagation. In line with previous studies [1, 25, 26], the branching network model illustrates how recurrent activity propagation leads to long timescales of single unit spiking activity, even though individual units in the branching network are memory-less (Fig 4B and 4C). Whereas recurrent activations clearly increase the timescale, the *magnitude* of single-unit correlation through recurrent activations is small [15]—and in fact much smaller than that induced by temporally correlated input (Fig 4D vs. Fig 4H). Hence, in the presence of correlated input (e.g., through visual stimulation with a natural movie), increasing recurrent amplification decreases predictability, because more and more spikes are caused by recurrent activations with lower predictability (Fig 4J). These model results are consistent with the experimental observation that predictability is higher under visual stimulation than for spontaneous activity, and positively correlated with stimulus selectivity (Fig 3C and 3F).

Yet, in the data, recurrent amplification might not be the only mechanism causing the observed long timescales. For example, clustered connectivity is known to lead to long timescales in balanced neural networks [90]. Overall, we do not expect this mechanisms to alter the effect of recurrence, but it might explain why timescales and predictability are higher in the data compared to the simple branching network. Another important mechanism besides recurrence that could give rise to a hierarchy of timescales is inter-areal feedback through long-range projections between cortical areas. For example, in a model based on measured connectivity in macaque cortex, long-range feedforward and feedback excitatory connections were shown to influence the exact layout of the hierarchy [1]. However, a more recent study found that a detailed balance between long-range excitatory inputs and local inhibitory inputs could prevent longer or shorter timescales to spill over from other areas, which yields a better segregation, and facilitates functional specialization [91]. Moreover, in [1], it was found that recurrence is a necessary requirement to achieve long timescales in the first place, since leaving out recurrent connections in the model led to much smaller timescales in general. This supports our perspective that recurrent amplification plays a key role in shaping temporal processing along the anatomical hierarchy.

However, evidence for stronger recurrent coupling in higher areas is not as clear in mice as it is in primates. In particular, the density and numbers of dendritic spines of layer 3 pyramidal neurons (which are interpreted as markers of recurrence [92]) are much more similar between

cortical areas in mice than in primates [42, 43, 45]. On the other hand, mouse cortex exhibits other gradients in microstructural properties [41], in particular a decrease in the density of parvalbumin (PV)-containing (inhibitory) interneurons for higher areas. This suggests that a gradient of recurrence in mice is rather controlled by the levels of inhibition than the net number (or density) of excitatory synapses [93].

## Coding perspective

Turning to a coding perspective, a hierarchy of timescales could also result from a feedforward representation of increasingly more complex visual features: A successive integration of temporally short-lived, simple features into temporally long-lived, complex ones could lead to long timescales in those neurons that represent the complex features [3, 8, 33, 71], which was found empirically in rat V1 [94]. In line with this, we found the hierarchy of information timescales to be more shallow under spontaneous activity (S12 and S13 Figs), indicating that visual stimulation has an effect on the information timescale hierarchy. However, we also found that median timescales and the correlation timescale hierarchy were very similar between visual stimulation and spontaneous activity, (Fig 3, S12 and S13 Figs), mirroring findings in monkey somatosensory cortex [95]. This indicates that stimulus encoding and complex feature representations do play a role, but cannot be the sole cause for long correlation and information timescales.

In particular, our results indicate that network-intrinsic mechanisms, such as an increase in recurrent coupling, are also important for shaping the temporal processing hierarchy. Such mechanisms are thought to cause stronger noise correlations in population codes [96], which were found to yield long timescales of information consistency and more stable representations for better readout and evidence accumulation in higher cortical areas [33, 97, 98].

However, feedforward feature representation and long timescales through intrinsic mechanisms are not necessarily separate processes. There exists the exciting possibility that the intrinsic generation of long timescales in neural activity could contribute to the activity-dependent learning of temporally invariant complex visual features [94, 99]. In this case, feedforward and recurrent processing would be inextricably linked for temporal processing, which could explain the similarity in median timescales across stimulus conditions.

Of particular interest from a coding perspective are also our results on predictability $R_{tot}$, because they imply that the temporal redundancy in individual spike trains decreases along the hierarchy. However, it is not clear which mechanism is responsible for such a decrease in redundancy. Our modeling results suggest that, in analogy to the timescale hierarchy, such a decrease in predictability can be caused by an increase in recurrent coupling, because stronger recurrent coupling can lead to a reduction of temporal correlations from external stimuli (Fig 4). Moreover, decreasing predictability could reflect efficient coding, where subsequent processing stages in cortex remove more and more temporal redundancy in individual spike trains [49, 79, 81]. On a mechanistic level, efficient-coding is in line with the observed increase in adaption along the cortical shape-processing hierarchy in rats [80], where adaptation attenuates responses to redundant, predictable stimuli [81, 82].

Finally, a decrease in predictability could be the result of complex, temporally invariant input representations formed in higher areas. These invariant representations have been shown to contain less stimulus information in absolute terms, because most of the information about low-level features such as contrast, luminance, position or phase is pruned in higher areas [76, 77], which also results in worse stimulus decoding performance for these areas [33]. Our results on predictability are compatible with this idea, because predictability is positively correlated with stimulus selectivity and higher under visual stimulation (Fig 3), hence a

decrease in predictability can be related to a decrease in absolute stimulus information for higher areas. Notably, all mechanisms may co-exist to serve complementary functions: Recurrence could enable long integration for temporal processing on the network level, while adaptation could ensure efficiency of the neural code on a single-neuron level, and both mechanisms could aid the formation of invariant, high-level representations by supporting temporal learning and the pruning of low-level, highly predictable information in higher cortical areas.

In comparison to cortex, thalamus displayed significantly lower predictability and timescales (Fig 2A–2C). However, whereas low timescales are in line with lower-level representations in thalamus, the fact that predictability is also lower than in all cortical areas suggests that additional mechanisms exist for temporal decorrelation, in line with redundancy reduction and efficient coding in LGN [62]. More generally, this might be due to the function of LGN and other thalamic areas as a relay of information, or a controller, requiring fast processing with little redundancy, little integration, and fast forgetting [63].

## Conclusion

Until now it has remained open whether single sensory pathways in cortex show a hierarchical organization of temporal processing, especially in rodents, where cortex exhibits less cognitive processing and fewer levels of abstractions compared to primates. Here, we found that the visual pathway in mouse cortex indeed shows signatures of such a hierarchy in the form of a gradual increase in correlation and information timescales, and a decrease in predictability along the anatomical hierarchy. Our analysis of data from different stimulus conditions and recurrent network models further indicated that the gradients in timescales are supported by stimulus-independent, network-intrinsic mechanisms, such as an increasing recurrence along the anatomical hierarchy. In contrast, the stimulus-dependent decrease in predictability is in agreement with efficient coding and the observed increase in adaptation in rodents, as well as a formation of increasingly invariant, high-level visual representations, and might constitute another hallmark of hierarchical temporal processing in mammals.

## Materials and methods

### Experimental data sets

To investigate temporal processing in the mouse visual system, we analyzed data from the *Visual Coding Neuropixels* data set, which is openly available through the Allen Brain Observatory [32, 57]. This data set contains extracellular electro-physiological recordings of mouse brain activity obtained with Neuropixels probes [100]. This setup enabled to simultaneously record from cortical and sub-cortical structures involved in visual processing (Fig 2A), with six cortical areas [primary visual cortex (V1), lateromedial area (LM), anterolateral area (AL), rostrolateral area (RL), anteromedial area (AM) and posteromedial area (PM)], and two thalamic areas [(the lateral geniculate nucleus (LGN) and lateral posterior nucleus (LP)], with a minimum of $n = 12$ and a maximum of $n = 24$ mice per brain area and experimental setup (see S1 Fig for the number of mice and analyzed sorted units per brain area). The probes record with high temporal precision with 30 kHz sampling rate and sub-millisecond temporal resolution, which is ideal to study temporally precise processing with spikes. For the experiments, the mice were head-fixed and were shown a range of visual stimuli.

The data set contains data from two experiments (*Functional Connectivity* and *Brain Observatory 1.1*), which differ in the sequences of stimuli that were shown to the mice. To study temporal processing in the mouse visual system in a naturalistic and stationary environment, our main results were obtained from the *Functional Connectivity* experiment, which contains two blocks

of around 15 minutes of consecutively recorded spiking activity during the repeated presentation of a 30 second naturalistic movie (termed `natural_movie_one_more_repeats` in the AllenSDK). Furthermore, we contrast our findings on the natural movie condition with results obtained from another single block of around 30 minutes of spontaneous activity, where the animal was shown a gray screen (termed `spontaneous`). In addition to the *Functional Connectivity* experiment, we also analyzed the *Brain Observatory 1.1* data, with two 10 minute blocks of natural movie stimulation with 120 second clips repeated 5 times per block (termed `natural_movie_three`). This did not only serve as a control, but also enabled the comparison of timescales and predictability to other metrics of visual processing that could be estimated within the *Brain Observatory 1.1* data set (Fig 3). To make results comparable across experiments and stimulus conditions, we adjusted the duration of the analyzed snippets of spike recordings to match the shorter recording duration of the natural movie condition of the *Brain Observatory 1.1*. Therefore, we only used the last 10 minutes of each block for the natural movie condition, and the last 20 minutes in the spontaneous activity condition in the *Functional Connectivity* data. This is important, because different recording lengths can cause systematic biases in the comparison of timescales and predictability [52]. Furthermore, at the beginning of each block a transient of 1 minute was removed to improve stationarity of the recording, and natural movie blocks were concatenated to then yield a total recording length of 18 minutes.

Finally, we only analyzed sorted units that fulfilled certain quality metrics and criteria that are relevant to our analysis. Spike sorting was done in [32], and we used the spike sorted data from there. Moreover, we only selected units based on quality metrics that are provided with the data [101]: a `presence_ratio_minimum` of 0.9 to filter out data corrupted by electrode drift, and an `isi_violations_maximum` of 0.5 to filter out units that contain inter-spike-intervals that violate a plausible refractory period. While these are the default values from the AllenSDK, we used an `amplitude_cutoff_maximum` of 0.01 (the AllenSDK default is 0.1) to ensure that most spikes are included. The number of sorted units after each filtering stage is shown in S1 Fig.

### Estimation of correlation timescale

The correlation timescale $\tau_C$ was estimated as the exponential decay rate of the autocorrelation single neuron spike trains [26, 32, 102]. Spike trains were obtained by binning spiking activity in bins of $\Delta t = 5$ ms, yielding a series of binary activations $a_t$, where $a_t = 0$ if there was no spike, and $a_t = 1$ if there was one or more spikes in the time interval $[t, t + \Delta t)$. The autocorrelation for time lags $T$ was then computed as

$$C(T) = \frac{\langle a_t a_{t-T} \rangle_t - \langle a_t \rangle_t^2}{\langle a_t^2 \rangle_t - \langle a_t \rangle_t^2},$$ (1)

where $\langle \cdot \rangle_t$ is the average over all times $t = T, T + \Delta t, \ldots, T_{rec} - \Delta t$, and $T_{rec}$ is the total recording time. From this, the correlation timescale $\tau_C$ was obtained by fitting an exponential decay $C(T) \propto \exp\left(-\frac{T}{\tau_C}\right)$ to the empirical autocorrelation function. Although [102] report that such a naive exponential fit can lead to biased estimates for *short* recordings, the long recording times $T_{rec}$ of roughly 20 minutes in the analyzed experiment were deemed sufficiently long for an unbiased estimation.

Moreover, the assumption of an exponentially decaying autocorrelation function is met for a branching process [61], and was found to be a good match for most of the neurons in the data (S9 Fig). However, we often observed an additional, slow decrease in the autocorrelation function for very long lags $T$, possibly reflecting scale-free fluctuations that might be induced

by sensory inputs with scale-free temporal correlations [103], or behavioral states [33]. To still estimate an correlation timescale $\tau_C$ we thus fitted the measured autocorrelation with a function consisting of two exponential terms

$$f(T) = A_1 \exp\left(-\frac{T}{\tau_1}\right) + A_2 \exp\left(-\frac{T}{\tau_2}\right). \tag{2}$$

Here, we defined the timescale with the larger coefficient $A_i$ as the intrinsic (dominant) timescale $\tau_C = \tau_i$, whereas the other coefficient $A_{j\neq i}$ corresponds to a secondary timescale $\tau_{\mathrm{sec}} = \tau_j$ that accounted for complementary effects like slow stimulus variables. Note that for $\approx 80\%$ of units $\tau_{sec}$ was larger than $\tau_C$, i.e., it mainly accounted for an additional, slow decay (see S6 Fig for a distribution of the fitted timescales). That slow stimulus variables have a smaller contribution to the autocorrelation is also in line with evidence for efficient coding, where neural representations were found to be strikingly invariant and adapt to slow stimulus variables through scale-free adaptation [104–106].

Fitting an additional decay made the analysis of the correlation timescale much more robust to the exact choice of the fitting range. In particular, the two-timescale fit yielded consistent estimates of $\tau_C$ when varying the maximum time lag $T_{\max}$ considered for fitting. In contrast, when fitting a single timescale function with offset $f(T) = A \exp\left(-\frac{T}{\tau_C}\right) + O$, the fitted $\tau_C$ increased monotonously with $T_{\max}$, because considering larger and larger lags $T$ biases estimates towards the slow (possibly scale-free) decay of the autocorrelation (S5 Fig). Yet, we found that considering these large time lags during fitting is crucial because a smaller fitting range (e.g., $T_{\min} = 500$ ms) biases estimates towards much shorter timescales (S5 Fig), and does not reveal a gradient of timescales along the anatomical hierarchy (S7 Fig). For the analysis of experimental data, we therefore used a two-timescale fit with $T_{\max} = 10$ s.

In addition, we found that the correlation timescale from a two-timescale fit was also less sensitive to deviations from an exponential decaying autocorrelation for small time lags. In particular, such deviations consisted of both, negative autocorrelation due to refractoriness and adaptation, or high initial autocorrelation, possibly indicating bursty firing or other short-term facilitating effects (S9 Fig). To exclude such short-term effects, we excluded autocorrelation coefficients $C(T)$ for short time lags $T < T_{\min}$ during the fit. Here, $T_{\min} = 30$ms was chosen, which gave the best fitting results for most neurons. An alternative approach to deal with such short-term effects is to fit an additional shorter timescale as in [26, 102], but this assumes an exponential decay for these short-term effects, which might not be the case in general. Moreover, whereas results from the single-timescale analysis were sensitive to the choice of $T_{\min}$, we found that results from the two-timescale fit did not crucially depend on $T_{\min}$ (S5 and S8 Figs). Thus, the overall impact of short-term effects on our results is comparably weak, which is also a result of the large fitting range (i.e., $T_{\max} = 10$ s) considered here.

To conclude, the hypothesis underlying this analysis is that between short and long timescales (in a range roughly between 20 to 1000 ms), there exists a decay that is characteristic for a unit's temporal processing, e.g., due to recurrent activity propagation in a network [15, 25, 26]. Here, we used a two-timescale fit that enables to extract such a characteristic timescale from the empirical autocorrelation function of individual units in a way that is robust to exact fitting parameters.

### Estimation of predictability and information timescale

To not only assess the timescale of time-lagged dependence, but also how predictable spiking is, we estimate a neuron's predictability as the relative spiking information $R(T)$ that can be

predicted from past spiking in a past range $T$ (Fig 1B) [52]. Since more past information can only increase predictability, $R(T)$ increases monotonously with $T$ until it reaches some value $R_{\text{tot}}$, the *total predictability* of a spike train. From this analysis we also obtain an *information timescale* $\tau_R$, which gives something similar to a rise time of the predictability, and indicates a typical timescale on which past information is informative, i.e. adds to the predictability of current spiking. Details on the measures and their estimation can be found in [52].

Here, we optimized $d = 5$ past bins for the estimation of predictability $R(T)$ for past ranges $T \in [T_{\text{min}}, T_{\text{max}}]$, with $T_{\text{min}} = 30$ ms and $T_{\text{max}} = 5$ s. The minimum past range of $T_{\text{min}} = 30$ ms was chosen so that the timescale $\tau_R$ only reflects past information *beyond* short-term effects due to a neuron's intrinsic spiking dynamics, similar to the exclusion of short time lags for the correlation timescale analysis (S10 Fig for a comparison of different $T_{\text{min}}$). For the maximum past range $T_{\text{max}} = 5$ s, we found that for all the analyzed units the estimated $R(T)$ reached the maximum value, which is required for the analysis of $R_{\text{tot}}$.

## Hierarchical Bayesian model for area differences

We used a hierarchical Bayesian analysis to investigate whether there are systematic area differences in the estimated timescales or predictability. This analysis enables to consider recordings for each mouse separately, while also incorporating effects that are shared among mice. The model assumes that the respective measure $y_{i,j} \in \{\tau_C, \tau_R, R_{\text{tot}}\}$ for each unit $i$ that was recorded in mouse $j$ is distributed according to a log-normal distribution, i.e. $\log Y_{i,j}$ is normally distributed

$$\log Y_{i,j} \sim \mathcal{N}(\mu_{i,j}, \epsilon). \tag{3}$$

Here, $\mu_{i,j}$ indicates the mean of the log measure (or $\exp(\mu_{i,j})$ the median of $Y_{i,j}$) and contains information about the unit $i$ and mouse $j$, whereas $\epsilon^2$ accounts for the unexplained variance. For the timescales we found that the distribution of log values was negatively skewed (S19 Fig), which cannot be accounted for by a normal distribution. Therefore we used a skew normal distribution for the timescales, with

$$\log Y_{i,j} \sim \text{SkewNormal}(\mu_{i,j}, \epsilon, \alpha), \tag{4}$$

where the additional shape parameter $\alpha$ controls the skewness of the distribution: $\alpha > 0$ gives a positively skewed, $\alpha < 0$ a negatively skewed, and $\alpha = 0$ recovers a normal distribution. An important property of this distribution is that, although mean and variance are not the same as for the normal distribution, the scale $\epsilon$ and shape $\alpha$ only add a constant to the mean. Therefore, differences in the log mean are still governed by $\mu_{i,j}$. For both models, the log mean

$$\mu_{i,j} = \theta_{\log v} \log v_{i,\text{norm}} + \theta_{\text{rf}} T_{\text{rf},i} + f_j(\text{area}_i) \tag{5}$$

is the sum of a predictor based on the unit's normalized log firing rate $\log v_{i,\text{norm}}$, a predictor $\theta_{\text{rf}}$ for visual responsiveness that is added if the unit has a significant receptive field on screen ($T_{\text{rf},i} = 1$, $T_{\text{rf},i} = 0$ otherwise), and a term $f_j(\text{area}_i)$ that models the dependence of the measure on the unit's area, and is specific for each mouse j (see below). Here, we included the firing rate and the existence of a receptive field in the model, because they were found to be correlated with some of the measures (S3 Fig), thus potentially improving the predictive power of the model. Moreover, not including them might lead to effects that are caused by, e.g., trivial differences in firing rate between different areas. Note that we use the logarithm of the firing rate $v$ divided by two standard deviations as regression input, so that its regression coefficient is comparable to the binary ones [107]. Hence, a difference of 1 in $\log v_{i,\text{norm}}$ (as between the two states of a binary variable) thus covers 2 standard deviations, e.g. the mean ±1 standard

deviation. As a prior, the standard normal distribution $\mathcal{N}(0, 1)$ was chosen for $\alpha$, $\theta_v$ and $\theta_{\text{rf}}$, whereas for the scale $\epsilon > 0$ a Half-Chauchy prior distribution HalfCauchy$(0, \beta)$ was chosen with scale parameter $\beta = 10$.

We used this approach to test (i) whether there are systematic differences in median time-scales and predictability between primary visual cortex (V1) and higher cortical areas (LM, AL, RL, AM, PM), and (ii) whether median timescales and predictability are linearly correlated with the anatomical hierarchy score of cortical areas, indicating a hierarchy of temporal processing. Note that because an area's median is related to the area's log mean via $\exp(\mu)$, the area specific term $f_j(\text{area}_i)$ is chosen as the logarithm of the assumed relation between an area and its median. Therefore, we built the following two models:

(i). A *cortical groups* model that incorporates an offset $\theta_{\text{hc},j}$ of the median for units from higher cortical areas, while the intercept $\theta_{0,j}$ governs the median for units from V1:

$$f_j(\text{area}_i) = \log(\theta_{0,j} + \theta_{\text{hc},j}\mathbf{1}_{\text{hc}}(\text{area}_i)), \tag{6}$$

(ii). A *cortical hierarchy* model that assumes a linear relation between an area's median and the anatomical hierarchy score of a unit's area HS($area_i$), with intercept $\theta_{0,j}$ and slope $\theta_{\text{hs},j}$:

$$f_j(\text{area}_i) = \log(\theta_{0,j} + \theta_{\text{hs},j}\text{HS}(\text{area}_i)). \tag{7}$$

Note that we shifted the hierarchy score to attain zero for V1 so that the intercept $\theta_{0,j}$ again reflects the median for units from V1.

All these parameters are modelled hierarchically, where for each mouse there is a parameter set $\boldsymbol{\theta}_j$, and each parameter $\theta_{k,j}$ of that set is drawn from a normal parent distribution with mean $\mu_{\theta_k}$ and variance $\mu_{\theta_k}$. This type of modelling respects that data are collected from different mice, but still allows to draw general conclusions based on the means $\mu_{\theta_k}$ of the parent distributions. As a prior for the means a standard normal distribution was assumed as a prior $\mu_{\theta_k} \sim \mathcal{N}(0, 1)$, whereas for the standard deviation a Half-Cauchy prior distribution was chosen $\sigma_{\theta_k} \sim \text{HalfCauchy}(0, \beta)$ with location 0 and scale parameter $\beta = 1$. We performed posterior predictive checks for both models and found that they were both well calibrated for the predictability and information timescale, whereas we found systematic deviations between the observed and modelled distribution of correlation timescales (S19 Fig). However, we do not expect this to have a significant effect on the present analysis, since the purpose is not to accurately describe the variability for individual data points, but to make statements about medians and average effects of specific predictors.

## Branching network model and simulations

To investigate how correlation timescale and predictability of single neurons depends on recurrent amplification, we employ a basic branching network that allows us to individually control external and recurrent activation. The network consists of $N = 1000$ binary units with state $s_i = \{0, 1\}$ on a random, sparse, directed, and weighted graph with mean degree $k = 10$ and connection weights $w_{ij}$ drawn randomly from $[0, 1)$. At discrete time steps ($\Delta t = 5$ ms), each unit can be activated ($s_i(t) \rightarrow 1$) recurrently and externally, and we define the population activity as $A(t) = \sum_i s_i(t)$. The probability to activate unit $i$ recurrently at time $t$ is

$$p_{i,\text{rec}}(t) = \sum_j w_{ij} s_j(t - 1). \tag{8}$$

To control the mean number of recurrent activations, and to ensure that each unit on average activates the same amount of other units, $w_{ij}$ is normalized such that $\sum_i w_{ij} = m$, where $m \in [0, 1)$ is the *neural efficacy*. Next to recurrent activations, the probability that unit $i$ is activated *externally* is given by a sigmoidal function

$$p_{i,\text{ext}}(t) = \left[1 + e^{-\frac{x_i(t)}{\sigma} - \gamma}\right]^{-1},\tag{9}$$

where $x_i(t) \in (-\infty, +\infty)$ is a time-dependent external drive, $\sigma$ sets the sensitivity to the external drive, and $\gamma$ is an offset adjusted to match the desired neural firing rate. To create temporally correlated input, we model $x_i(t)$ as an Ornstein–Uhlenbeck process with a timescale $\tau_{\text{ext}} = 30$ ms; to create uncorrelated input, we set $x_i(t) = 0$. To match the average neural firing rate to the experimentally observed value $v^* \approx 3.5$ Hz (which corresponds to a population activity of $A^* = Nv^* \Delta t$), we initialize $\gamma = \ln\left(\frac{1-ma}{1-a} - 1\right)$ (the mean-field solution for $x_i(t) = 0$) and homeostatically regulate a global $\gamma$ parameter in each time step as

$$\tau_\gamma \Delta\gamma = \Delta t[A^* - A(t)]/N,\tag{10}$$

where $\tau_\gamma = 60$ s is a slow time scale such that $\gamma$ changes very little during the recording.

To estimate the mean and statistical error of correlation timescales and predictability, we generate 20 random network realizations for each $m$. For each realization, the simulation is first equilibrated for 20 minutes to ensure stationary dynamics, before we record for another 20 minutes spiking activity from $n = 20$ randomly selected neurons. From the spiking activity, $\tau_C$, $\tau_R$ and $R_{\text{tot}}$ are calculated for each neuron using the same tools as for the experimental data. The only differences are that (i) we removed the lower bound of the fit ranges (starting at $T_{\min} = 1$ time step) and (ii) for fitting $C(T)$ we used a simple exponential with offset (only featuring one timescale) since no separate fast and slow timescales needed to be distinguished in this simple model (see Discussion).

## Supporting information

**S1 Fig. Number of available units for the analysis for the *Functional Connectivity* data set.** (**A**) Sorted units available for analysis after spike sorting process ("valid wave-forms"), after applying filters of the AllenSDK ("quality metrics"), after selecting only units of the "Functional Connectivity set" and after ensuring that the recordings of the selected "stimuli" for each unit are long enough and do not include invalid spike times (Materials and methods). (**B**) Numbers of units for each session and for each area available for analysis after filtering. (PDF)

**S2 Fig. Relation between correlation and information timescales, as well as predictability across all sorted units.** Histograms of the correlation timescale $\tau_C$, the information timescale $\tau_R$ and the predictability $R_{\text{tot}}$ (diagonal), as well as scatter plots of one measure against the other (y-axes refer to scatter plots, no axes shown for histograms) for all analyzed units. Scatter plots are overlaid with kernel density estimations, where lines indicate regions of equal probability. Correlation and information timescales are shown in log scale. Timescales are positively correlated (Pearson correlation), whereas predictability is weakly negatively correlated with the timescales. (PDF)

**S3 Fig. Relation of correlation and information timescales, as well as predictability to common firing statistics.** Scatter plots of a measure of timescale or predictability versus common firing statistics such as the average firing rate, median inter-spike-interval (ISI) and coefficient of variation (CV) for all analyzed units. Scatter plots are overlaid with kernel density

estimations, where lines indicate regions of equal probability. Correlation and information timescales, as well as median ISIs are shown in log scale. The firing rate is mainly positively correlated with the correlation timescale, and negatively correlated with the predictability (Pearson correlation). The median ISI is mainly correlated with the information timescale, whereas the CV is strongly correlated with the predictability.
(PDF)

**S4 Fig. Correlation between single neuron firing statistics and hierarchy score for natural movie stimulation in the *Functional Connectivity* data set.** Firing rate and median inter-spike-interval (ISI) of sorted units tend to increase with hierarchy score, but have lower Pearson correlation coefficients $r_P$ (with higher p-values $P_P$), as well as Spearman correlation coefficients $r_S$, compared to timescales or predictability on the same data set in Fig 2. The coefficient of variation (CV), in contrast, is not correlated with the hierarchy score.
(PDF)

**S5 Fig. For single timescale fit, median correlation timescale and hierarchy score correlation are sensitive to the maximum time lag used for fitting. (A)** For a single timescale fit, the median estimated timescale (over all cortical units) increases monotonously with maximum lag $T_{\max}$, independent of the minimum time lag $T_{\min}$ used for fitting (light and dark blue lines). In contrast, when using a two-timescale fit, the median estimated correlation timescale remains consistent for sufficiently large max lags $T_{\max} > 1000$ ms, for both, $T_{\min} = 5$ ms and $T_{\min} = 30$ ms (light and dark green lines). However, timescales are larger for $T_{\min} = 5$ ms, because often fits are flattened by negative autocorrelation for short time lags, which are mostly excluded with $T_{\min} = 30$ ms. **(B)** The Pearson correlation coefficient $r_P$ between an area's median correlation timescale and anatomical hierarchy score (c.f. Fig 2D) is consistently high if $T_{\max}$ is chosen sufficiently high. **(C)** Similarly, for sufficiently large $T_{\max}$, the p-value of the fit is consistently below 0.05. For all plots, red dot indicates the $T_{\max} = 10$s used for the main analyses. Timescales were estimated for the natural movie condition of the *Functional Connectivity* data set.
(PDF)

**S6 Fig. Distribution of fitted timescales for the single and two-timescales fit.** To assess how estimated correlation timescales differ for the single and two-timescale fits, we here show the distribution of fitted timescales for the single timescale fit (blue), and the two fitted timescales for the two-timescale fit (orange). For the latter, we show both, the selected timescale (see Materials and methods), which yields the estimate of the correlation timescale, and the rejected timescale. The single timescale is generally larger than the selected timescale of the two-timescale fit, because it also accounts for a potential slow decay of autocorrelation on long timescales. For the two-timescale fit, in contrast, it is mostly the rejected timescale that accounts for the long timescales, since the rejected timescales are generally much larger.
(PDF)

**S7 Fig. No hierarchy of correlation timescales is found for a small fitting range.** To demonstrate the effect of the fitting range on the inferred hierarchy of correlation timescales $\tau_C$, we repeated the same analysis from Fig 2D for a smaller maximum time lag $T_{\max} = 500$ ms, and different minimum time lags $T_{\min}$. For this choice of $T_{\max}$, no hierarchy is found, and median values of $\tau_C$ are in general much lower (dots, bars indicate bootstrapping confidence intervals on median). This is found for all choices of $T_{\min}$, indicating that this is primarily caused by omitting larger time lags $T > T_{\max}$ during fitting. Here, correlation timescales were computed for spiking activity under natural movie stimulation in the *Functional Connectivity* data set. Moreover, $\tau_C$ was obtained from a single timescale fit to enable a comparison to previous

analyses (c.f. Extended Data Fig 9 in [32]), but we obtained a similar result also for the two-timescale fit, although this approach is generally much more robust to the choice of fitting range (c.f. S5 Fig).
(PDF)

**S8 Fig. For a sufficiently large fitting range, the hierarchy of timescales is found independent of the exclusion of small time lags during fitting.** For a large fitting range with $T_{max}$ = 10 s, we consistently find a hierarchy of correlation timescales $\tau_C$, whereas $T_{min}$ only slightly affects the exact layout of the hierarchy, and the median values of $\tau_C$ (c.f. S5 Fig). Here, correlation timescales were computed for spiking activity under natural movie stimulation in the *Functional Connectivity* data set. Moreover, timescales where obtained using the two-timescale fitting procedure (Materials and methods), because this is the analysis used for all main results obtained in this paper.
(PDF)

**S9 Fig. Randomly selected examples of autocorrelation functions and single and two-timescale fits for different fitting ranges.** Measured autocorrelation functions (grey line) for units in the *Functional Connectivity* data set under natural movie stimulation. Black dots and green lines indicate single and two-timescale fits, respectively, with the inferred timescale stated in the corresponding color.
(PDF)

**S10 Fig. Hierarchy of information timescale relies on excluding small past ranges from the analysis.** Similar to the correlation timescale, we excluded past ranges smaller than some $T_{min}$ when computing the information timescale $\tau_R$ to exclude short-term effects like refractoriness and tonic firing (Materials and methods). Here, we show median information timescales for cortical areas, each time for a different choice of minimal past range $T_{min}$. Notably, differences between higher cortical areas (LM, AL, PM, AM) are only visible when excluding past ranges smaller than $T_{min}$ = 30 ms. Information timescales were computed for spiking activity under natural movie stimulation in the *Functional Connectivity* data set.
(PDF)

**S11 Fig. Timescale and predictability of different brain areas under natural movie stimulation in the *Brain Observatory 1.1* data set. (A–C)** As for recorded activity in the *Functional Connectivity* data set, the medians of all measures differ significantly for different structural groups (thalamus, primary visual cortex and higher cortical), with the same ordering as before. Black boxes indicate the median over sorted units of the different structural groups, whereas coloured dots indicate the median for individual areas. Bars indicate standard deviation on the median obtained from bootstrapping. **(D–F)** Measures across the cortical hierarchy show the same general increase as for the *Functional Connectivity* data set. However, median correlation and information timescales show higher variability, and thus we find smaller correlation coefficients (dashed line, Pearson and Spearman correlation coefficients and corresponding p-values shown below).
(PDF)

**S12 Fig. Timescales and predictability of different brain areas for *spontaneous activity* in the *Functional Connectivity* data set. (A–C)** As for recorded activity under natural stimulation, the medians of correlation and information timescales differ significantly between different structural groups (thalamus, primary visual cortex and higher cortical), with the same ordering as before. However, the difference in median predictability between V1 and higher cortical areas is not significant for spontaneous activity. Black boxes indicate the median over

sorted units of the different structural groups, whereas coloured dots indicate the median for individual areas. Bars indicate standard deviation on the median obtained from bootstrapping. **(D–F)** Measures across the cortical hierarchy show the same general trend as for natural movie stimulation. However, in general, measures for different areas are more similar, leading to smaller correlation values with the hierarchy score and higher p-values (dashed line, Pearson and Spearman correlation coefficients and corresponding p-values shown below).
(PDF)

**S13 Fig. Posterior distributions of the hierarchy score slope reveal a significant increase in timescales and decrease in predictability with hierarchy score.** To assess whether timescales and predictability relate to the anatomical cortical hierarchy, a linear relationship between an area's median and anatomical hierarchy score was modelled with slope $\theta_{hs}$. **(A)** (Top) For the correlation timescale, the 95% posterior credible interval of the mean hierarchy score slope $\mu_{\theta_{hs}}$ across all mice is positive (black bar, red dot indicates median), indicating that there is an increase in median correlation timescales with the hierarchy score. (Bottom) On the level of individual mice, posteriors indicate the same effect, but are more diverse (colors indicate different mice). In particular, for some mice the posteriors also attribute probability to zero or negative slopes, which could be either due to increased uncertainty due to the smaller sampling size, or an incomplete sampling of the areas for individual mice. **(B)** For the information timescale $\tau_R$, the posterior credible interval of the mean slope is also positive. **(C)** For predictability, in contrast, the credible interval is negative, indicating a credible decrease in predictability with hierarchy score. **(D–F)** Very similar results are obtained for the *Brain Observatory* data set. **(G)** For spontaneous activity and the correlation timescale, the posterior of the mean slope is very similar to the natural movie conditions, but variability across mice is larger. **(H)** For the information timescale, in contrast, the posterior indicates a smaller slope. **(I)** For predictability, the credible interval even contains a zero slope, indicating that predictability does not necessarily decrease along the anatomical hierarchy for spontaneous activity.
(PDF)

**S14 Fig. Posterior distributions of intercepts in the cortical hierarchy model.** Overall, posteriors over intercepts of the cortical hierarchy model reveal a high diversity among mice, whereas slopes are more similar between different mice (S13 Fig). **(A)** (Top) Posterior distributions of the mean $\mu_{\theta_0}$ and standard deviation $\sigma_{\theta_0}$ of the model intercept $\theta_0$ for the correlation timescale. (Bottom) Posterior distributions of $\theta_0$ for individual mice (colors indicate different mice). **(B,C)** Same as A, but for information timescale and predictability. **(D–F)** Same as A–C, but for the *Brain Observatory* data set. **(G–I)** Same as A–C, but for spontaneous activity in the Functional Connectivity data set. **(I)** For predictability, intercepts are much more similar between mice, because the slope is much closer to 0 (S13 Fig), and thus may partially account for differences in median predictability between mice. For all panels red dots and black bars indicate the median and 95% highest-density-interval of the posterior distribution.
(PDF)

**S15 Fig. Posterior distributions of the higher cortical offset reveal a significant increase in timescales and decrease in predictability for higher cortical areas.** To assess the difference in temporal processing between higher cortical areas and V1, the median timescales and predictability in higher cortical areas were modelled with an offset $\theta_{hc}$. **(A)** (Top) For the correlation timescale, the 95% posterior credible interval of the mean offset $\mu_{\theta_{hc}}$ across all mice is positive (black bar, red dot indicates median), indicating a credible increase in timescales for higher cortical areas. (Bottom) On the level of individual mice, posteriors indicate the same effect, but are more diverse (colors indicate different mice). In particular, for some mice the

posteriors also attribute probability to zero or negative offsets, which could be either due to increased uncertainty due to the smaller sampling size, or an incomplete sampling of the areas for individual mice. **(B)** For the information timescale the posterior credible interval of the mean offset is also positive. **(C)** For predictability, in contrast, the credible interval is negative, indicating a credible decrease in predictability for higher cortical areas. **(D–F)** Very similar results are obtained for the *Brain Observatory* data set. **(G)** For spontaneous activity and the correlation timescale, the posterior of the mean offset is very similar to the natural movie conditions, but variability across mice is larger. **(H)** For the information timescale, in contrast, the posterior indicates a smaller offset. **(I)** For predictability the credible interval even contains a zero offset, indicating that predictability under spontaneous activity is not necessarily smaller in higher cortical areas when compared to V1.
(PDF)

**S16 Fig. Posterior distributions of intercepts in the cortical groups model.** Overall, posteriors over intercepts of the cortical groups model reveal a high diversity among mice, whereas offsets are more similar between different mice (S15 Fig). **(A)** (Top) Posterior distributions of the mean $\mu_{\theta_0}$ and standard deviation $\sigma_{\theta_0}$ of the model intercept $\theta_0$ for the correlation timescale. (Bottom) Posterior distributions of $\theta_0$ for individual mice (colors indicate different mice). **(B, C)** Same as A, but for information timescale and predictability. **(D–F)** Same as A–C, but for the *Brain Observatory* data set. **(G–I)** Same as A–C, but for spontaneous activity in the *Functional Connectivity* data set. **(I)** For predictability, intercepts are much more similar between mice, because the higher cortical offset is much closer to 0 (S15 Fig), and thus may partially account for differences in median predictability between mice. For all panels red dots and black bars indicate the median and 95% highest-density-interval of the posterior distribution.
(PDF)

**S17 Fig. Posteriors for non-hierarchical parameters of the cortical hierarchy model. (A)** Posterior density function of non-hierarchical parameters for the *Functional Connectivity* data set (red dot indicates median and black bar the 95% highest density interval). The highest density interval (HDI) for the receptive field predictor $\theta_{rf}$ is negative for the timescales for all stimulus conditions (A–C), hence units with a significant receptive field tend to have smaller timescales. In contrast, the HDI is positive for predictability for the movie conditions (A,B) and negative for spontaneous activity (C), indicating that units with a receptive field have higher predictability when driven with a stimulus with temporal correlations, in line with the idea that predictability is induced by visual stimuli. The HDI for the log firing rate predictor $\theta_{\log \nu}$ is positive for the correlation timescale and negative for the information measures, thus units with higher firing rate tend to have longer correlation timescales but smaller predictability. The posterior of the shape parameter $\alpha$ is concentrated on negative values, indicating a negatively skewed distribution of log correlation and information timescales. **(B, C)** Apart from posteriors of the receptive field predictor for predictability, the posteriors for the natural movie condition in the *Brain Observatory 1.1* data set (B) and spontaneous activity in the *Functional Connectivity* data set (C) are overall very similar to the posteriors in (A).
(PDF)

**S18 Fig. Posteriors for non-hierarchical parameters of the cortical groups model.** Posterior densities for non-hierarchical parameters of the cortical groups model are very similar to the hierarchy score model (c.f. S17 Fig).
(PDF)

**S19 Fig. Posterior predictive checks of the different Bayesian models applied to the *natural movie* condition in the *Functional Connectivity* data set. (A)** To test whether the cortical

hierarchy model is well calibrated, we compared the posterior predictive likelihood of the model (blue line) with the observed likelihood of timescales and predictability (black line), which overall show a good agreement. Note the skewness in the distribution of log timescales, which led us to use a skew normal distribution to model the residual variability in the timescale data. To obtain a better check for the conditional probabilities of individual data points, and not only the pooled data, we performed LOO cross-validated probability integral transform (PIT) posterior predictive checks [108]. In LOO-PIT, the model is fitted for each datum $y_i$ to all data except $y_i$, here denoted as $\mathbf{y}^{-i}$. $\Pr(y_i^{\mathrm{model}} \leq y_i|\mathbf{y}^{-i})$ then represents the probability that a value $y_i^{\mathrm{model}}$ simulated from the fitted model is less or equal to $y_i$. If the model and data distributions are the same, then the distribution of these probabilities over all data points $y_i$ (thick blue line) should be uniform [108], hence we compare it to 100 simulated data sets from a uniform distribution (thin blue lines). The model appears well calibrated for the predictability, whereas for the timescales the model tends to under-represent intermediate values, and to over-represent small and high values. **(B)** Same as A, but for the cortical groups model. Notably, the models appear to be equally well calibrated, hence we do not expect the differences in their predictive power to be caused by a sub-optimal calibration of the models.
(PDF)

**S20 Fig. Posterior predictive checks of the different hierarchical models applied to the natural movie condition in the *Brain Observatory* data set.** Same as S19 Fig, but for the natural movie condition in the *Brain Observatory 1.1* data set. For these data, the model is slightly worse calibrated for the correlation timescale.
(PDF)

**S21 Fig. Posterior predictive checks of the different hierarchical models applied to *spontaneous activity* in the *Functional Connectivity* data set.** Same as S19 Fig, but for spontaneous activity in the *Functional Connectivity* data set. For these data, the model is slightly worse calibrated, in particular for the correlation timescale and predictability.
(PDF)

**S22 Fig. Area-wise comparison of timescales and predictability between natural movie and spontaneous activity in the *Functional Connectivity* data set.** We compared across areas to determine if there was a systematic difference in timescales and predictability between stimulation with a natural movie and spontaneous activity under grey screen illumination. While median correlation timescales do not systematically differ between stimulus conditions, the median information timescale is significantly larger under spontaneous activity for most areas, whereas median predictability is significantly smaller (p-values were obtained using Wilcoxon signed-rank tests, where only significant p-values after Bonferroni correction are reported). Moreover, the relative median difference Δ between conditions is higher for predictability, indicating a stronger overall effect.
(PDF)

**S23 Fig. Relation of timescales and predictability to direction selectivity for different visual areas.** For some cortical visual areas the direction selectivity of individual units (measured on drifting gratings shown in 8 different directions [32]) is negatively correlated with the correlation and information timescale and for all areas positively correlated with predictability. Dots show values for each unit and lines show the linear regression with Pearson correlation coefficient $r$ and corresponding two sided p-value $p$. Regression lines are only shown for areas with significant correlations after Bonferroni multiple comparison correction. For cortical areas V1 and LM, information timescales are also negatively correlated with direction

selectivity.
(PDF)

**S24 Fig. Relation of timescales and predictability to image selectivity of individual units for all cortical areas. (A)** Correlation timescales $\tau_C$ of individual cortical units (dots) under stimulation with a natural movie in the *Brain Observatory 1.1* data set versus the image selectivity index (measured for different static images shown to the mice [32]). Timescales $\tau_C$ show a weak negative correlation with image selectivity index (line shows linear regression, *r* gives Pearson correlation coefficient with corresponding two sided p-value). **(B)** Information timescales $\tau_R$ are very weakly negatively correlated with image selectivity index. **(C)** In contrast, the predictability $R_{tot}$ is positively correlated with the image selectivity index.
(PDF)

**S25 Fig. Relation of timescales and predictability to image selectivity for different visual areas.** For cortical visual areas the image selectivity of individual units (measured for different static images shown to the mice [32]) is negatively correlated with the correlation timescale and positively correlated with predictability. For some areas, also information timescales are negatively correlated with image selectivity. Dots show values for each unit and lines show the linear regression with Pearson correlation coefficient *r* and corresponding two sided p-value *p*. Regression lines are only shown for areas with significant correlations after Bonferroni multiple comparison correction.
(PDF)

**S26 Fig. Timescales and predictability in a plastic recurrent LIF network implemented on neuromorphic hardware.** To compare results from the branching network with a more realistic recurrent network, we consider an implementation of a network of $N = 512$ leaky integrate-and-fire neurons with synaptic plasticity on neuromorphic hardware (BrainScaleS-2) [63, 75]. In this implementation, the recurrent amplification (expressed via neural efficacy $m=1-h/a$) cannot be set directly, but is tuned via plasticity, which adapts to the number of input synapses $k_{in}$ from which each unit receives Poisson input. In particular, it has been shown that for less external input, the spike-timing-dependent plasticity tunes the network towards configurations with stronger recurrent coupling, and better integration for complex tasks [63]. To quantify the effective strength of recurrent coupling, we estimated the neural efficacy $m$ via autoregression of the activity time series. As in the branching network, an increase in recurrence (here expressed through $m$, and shown for a smaller range) increases $\tau_C$ and $\tau_R$, but decreases $R_{tot}$. Notably, in this model the source of single-neuron predictability besides recurrence is not provided through temporal correlations in the input, but by the membrane dynamics, effectively causing single-unit memory and predictability, which then gets diminished by increasing recurrence. Small dots show median values for individual network realizations, and big dots indicate median values over all network realizations for a given $k_{in}$.
(PDF)

## Acknowledgments

We want to thank Davide Zoccolan, Jorge Jaramillo, Brandon Munn and the Priesemann group, especially Jonas Dehning, Fabian Mikulasch and Andreas Schneider, for helpful discussions and comments on the manuscript.

## Author Contributions

**Conceptualization:** Lucas Rudelt, Viola Priesemann.

**Formal analysis:** Lucas Rudelt, Daniel González Marx, F. Paul Spitzner.

**Funding acquisition:** Viola Priesemann.

**Investigation:** Lucas Rudelt, Daniel González Marx.

**Methodology:** Lucas Rudelt, Daniel González Marx, Benjamin Cramer, Johannes Zierenberg, Viola Priesemann.

**Project administration:** Viola Priesemann.

**Software:** Lucas Rudelt, Daniel González Marx, F. Paul Spitzner, Benjamin Cramer, Johannes Zierenberg.

**Supervision:** Viola Priesemann.

**Validation:** Lucas Rudelt.

**Visualization:** Lucas Rudelt, Daniel González Marx, F. Paul Spitzner.

**Writing – original draft:** Lucas Rudelt, F. Paul Spitzner.

**Writing – review & editing:** Lucas Rudelt, F. Paul Spitzner, Benjamin Cramer, Johannes Zierenberg, Viola Priesemann.

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
