## [Decision Letter · Decision Letter 0]

4 Mar 2024

Dear Mr. Rudelt,

Thank you very much for submitting your manuscript "Signatures of hierarchical temporal processing in the mouse visual system" for consideration at PLOS Computational Biology. As with all papers reviewed by the journal, your manuscript was reviewed by members of the editorial board and by several independent reviewers. The reviewers appreciated the attention to an important topic. Based on the reviews, we are likely to accept this manuscript for publication, providing that you modify the manuscript according to the review recommendations.

Sincerely,

Daniele Marinazzo

Section Editor

PLOS Computational Biology

Daniele Marinazzo

Section Editor

PLOS Computational Biology

Reviewer's Responses to Questions

**Comments to the Authors:**

Reviewer #1: I liked this paper a lot, and I believe that the comparison of intrinsic vs coding time scales is of high importance.

I have no concerns about the analyses and interpretations, which are done to a high standard.

There are aspects about background literature and interpretations of results that would deserve a better elaboration and a little more thinking, to make sure the implications of the results are fully exploited and better placed in the context of the existing literature.

The first relates to the relationship between timescales and efficient coding. There are a couple of papers that seems highly relevant and should be cited and discussed in this context. Młynarski and Hermundstad Nature Neuroscience 2021. Fairhall et al Nature 2001.

The second is related to the temporal redundancy and its possible functional role. The discussion of this seems limited to the role that long timescales can have in predictability. An independent view of what could be the functional role of these timescales (which as the authors point out in the Introduction, can be due to coupling between cells not only to single-cell mechanisms), is that the benefits of long timescales for perceptual discriminations so far have been strongly imputed to the fact that these longer information timescales facilitate the downstream readout of information, relating to the fact that these timescales are due to or accompanied by stronger time-delayed coupling between cells. See Valente et al, Nature Neuroscience 2021. Panzeri et al Nature Reviews Neuroscience 2022. This evidence should be elaborated in the Discussion section about long information timescales.

Stefano Panzeri

Reviewer #2: I uploaded my review as an attachment

**Have the authors made all data and (if applicable) computational code underlying the findings in their manuscript fully available?**

Reviewer #1: Yes

Reviewer #2: Yes

PLOS authors have the option to publish the peer review history of their article (what does this mean?). If published, this will include your full peer review and any attached files.

Reviewer #1: **Yes: **Stefano Panzeri

Reviewer #2: **Yes: **Davide Zoccolan

Figure Files:

Data Requirements:

Reproducibility:

References:

---

## [Editor Report · Decision Letter 1]

23 Jul 2024

Dear Mr. Rudelt,

We are pleased to inform you that your manuscript 'Signatures of hierarchical temporal processing in the mouse visual system' has been provisionally accepted for publication in PLOS Computational Biology.

Best regards,

Daniele Marinazzo

Section Editor

PLOS Computational Biology

Daniele Marinazzo

Section Editor

PLOS Computational Biology

---

## [Editor Report · Acceptance letter]

15 Aug 2024

PCOMPBIOL-D-24-00096R1 

Signatures of hierarchical temporal processing in the mouse visual system

Dear Dr Rudelt,

I am pleased to inform you that your manuscript has been formally accepted for publication in PLOS Computational Biology. Your manuscript is now with our production department and you will be notified of the publication date in due course.

With kind regards,

Anita Estes
